# CLUE-NAS: A CLIP-Inspired Contrastive Learnable Unifying Encoder for Neural Architecture Search

## Abstract

Conventional encoder-based neural architecture search (NAS) methods typically encode candidate architectures as graphs based on their information flow and operations. Such graph-based embeddings primarily capture topological features, such as nodes and edges, while lacking high-level semantic representations, which limits the robustness and generalization of encoder-based NAS. This issue is evident in several phenomena, such as the inability of typical NAS methods to interpret previously unseen operations or their limited capacity to benefit from joint training across multiple search spaces. To mitigate these limitations, we propose Contrastive Learnable Unifying Encoder for NAS (CLUE-NAS), a novel framework that leverages the text encoder of Contrastive Language Image Pre-training (CLIP) to generate context embeddings enriched with high-level semantics and integrates them with graph-based embeddings through contrastive learning. CLUE-NAS further emulates human expert behaviors by employing a coarse-to-fine strategy to enhance performance. Experiments on NASBench-101, NASBench-201, and NASBench-301 show that CLUE-NAS not only demonstrates strong generalization to unseen operations but also benefits substantially from joint training, achieving competitive results against state-of-the-art NAS baselines.

## 1 Introduction

Neural Architecture Search (NAS) aims to discover optimal architecture within a predefined search space containing numerous candidates. Early NAS methods based on reinforcement learning Zoph & Le (2016); Baker et al. (2016); Tan et al. (2019) incurred prohibitively high search costs, while differentiable approaches such as DARTS Dong & Yang (2019); Liu et al. (2018b); Cai et al. (2018); Xie et al. (2018) suffered from instability when transitioning from continuous to discrete architectures. Moreover, these methods often require substantial computational resources. To address these limitations, a range of low-cost alternatives Ru et al. (2021); Abdelfattah et al. (2021); Lin et al. (2021); Li et al. (2023) have emerged, including NTK-based methods Chen et al. (2021); Xu et al. (2021); Zhu et al. (2022). However, such methods typically rely on complex theoretical assumptions that often fail to hold in practice. For instance, both LGA Mok et al. (2022) and MOTE Zhang et al. (2024b) have shown that the assumptions underlying NTK, which are derived under the premise of infinitely wide networks, do not generalize well to real-world architectures with finite width, resulting in poor predictive performance.

Another promising direction is the encoder-based NAS framework Liu et al. (2018a); Luo et al. (2018); Wen et al. (2020); Dudziak et al. (2020); Wei et al. (2022); Wu et al. (2021); Huang et al. (2022); Zhang et al. (2023), which strikes a more favorable balance between efficiency and accuracy. These approaches often outperform DARTS-based methods in speed and surpass NTK-based methods in predictive performance. However, their effectiveness largely depends on the architecture representation quality. A common strategy is to encode architectures as graphs, where nodes represent operations and edges represent information flow. While effective to some extent, this graph-based encoding approach typically relies on one-hot vectors to represent operations. Unfortunately, this labeling method is too hard to generalize across search spaces, as each benchmark. For example, NASBench-101 Ying et al. (2019), NASBench-201 Dong & Yang (2020), and NASBench-301 Siems et al. (2020) define their own distinct sets of operations. For instance, the standard

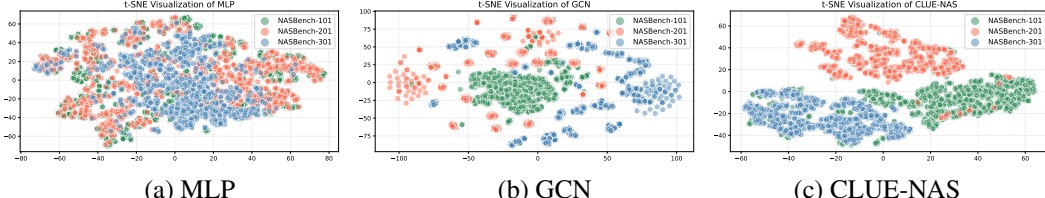

Figure 1: The three figures illustrate the distribution of architecture embeddings obtained using different encoders. The left figure corresponds to MLP, the middle to GCN, and the right to the proposed CLUE-NAS. All three encoders were trained on NASBench-101 and evaluated on NASBench-101, NASBench-201, and NASBench-301, respectively. Notably, only CLUE-NAS demonstrates a clear ability to generalize to unseen architectures from NASBench-201 and NASBench-301. In contrast, both MLP and GCN fail to capture meaningful representations for these unseen architectures.

convolutions used in NASBench-101 and NASBench-201 differ significantly from the separable convolutions used in NASBench-301. Consequently, one-hot encoding fails to capture cross-space generalizability. Fig. 1 shows some evidence about this issue, which the trained MLP and GCN on NASBench-101 cannot identify the architectures of NASBench-201 and NASBench-301. For a complete discussion of Fig. 1, please refer to the Appendix B.

Moreover, these graph-based encoders primarily focus on topological structure, overlooking the semantic cues that human experts often rely on when evaluating architectures. In contrast, human experts rarely assess neural architectures based solely on their wiring diagrams. Instead, they draw from experience to make intuitive and qualitative assessments such as "this model looks heavy" or "this one seems efficient but shallow." These judgments stem not from explicit numerical metrics or topological analysis but from flexible and approximate semantic reasoning. This human intuition enables robust performance estimation across a wide range of architectures, from VGG Simonyan & Zisserman (2014) and ResNet He et al. (2016) to MobileNet Howard et al. (2017); Sandler et al. (2018) and Vision Transformers Dosovitskiy et al. (2020). In short, existing encoder-based NAS frameworks treat performance prediction as a regression task over graph embeddings, rendering them topology-aware but semantically blind, which hampers their generalization and robustness.

The motivation behind this paper is twofold. First, while recent advances in neural architecture search (NAS) have benefited from predictive encoders and zero-cost proxies, they remain largely confined to topology-based representations that treat architectures as graphs with discrete, hard-labeled operations. Such encodings are limited in expressiveness, difficult to transfer across heterogeneous search spaces, and fail to capture higher-level design intent. Second, there is a growing gap between how human experts reason about architectures—often using intuitive, semantic judgments, like 'deep,' 'sparse,' or 'transformer-like', and how machine-based encoders currently operate. Bridging this gap calls for a richer, more flexible representation that integrates structural and semantic information. In response, we propose a CLIP-inspired contrastive learning framework that unifies graph-based and language-based architectural descriptions, enabling performance prediction that is both generalizable and semantically grounded.

Inspired by the way human experts assess neural architectures, we propose CLUE-NAS (Contrastive Learnable Unifying Encoder for Neural Architecture Search) to generate context embeddings with high-level semantics and integrate them with graph-based embeddings through contrastive learning. Unlike prior approaches that rely solely on topological embeddings, CLUE-NAS jointly aligns and integrates topological representations with context embeddings extracted from CLIP's text encoder, which capture high-level semantic information. Furthermore, CLUE-NAS emulates human reasoning by combining semantic priors with fine-grained architectural analysis to predict the performance of architectures in a coarse-to-fine manner. This fusion of semantic and structural understanding significantly enhances the predictive accuracy and generalizability of NAS models, as demonstrated by our experimental results. CLUE-NAS consistently outperforms baseline methods on NASBench-101, NASBench-201, and NASBench-301. Our key contributions are summarized as follows.

- **Novel dual-view representation of architectures,** combining graph-based structural characteristics (operation metrics and information flow) with semantic context embeddings derived from descriptions of natural languages using the CLIP text encoder.

- **Contrastive context alignment mechanism,** which aligns graph-based embeddings with context embeddings in a coarse-to-fine manner. This mimics human evaluation behavior and enhances the robustness of representation learning.
- **Human-like confidence prediction strategy,** where a coarse semantic estimate is refined through a learned offset, providing accurate and bounded performance prediction for candidate architectures.
- **Strong empirical results across multiple NAS benchmarks,** demonstrating that CLUE-NAS significantly outperforms existing encoder-based and zero-cost NAS in both accuracy and generalizability, while maintaining low computational overhead.

## 2 RELATED WORK

Neural Architecture Search (NAS) has been extensively studied, with early approaches relying on reinforcement learning Zoph & Le (2016); Baker et al. (2016); Tan et al. (2019) and differentiable frameworks Dong & Yang (2019); Liu et al. (2018b); Cai et al. (2018); Xie et al. (2018). Some methods employ evolutionary algorithms Lu et al. (2019); Real et al. (2019; 2017); Xie & Yuille (2017); Dai et al. (2021) or RNN-based strategies to iteratively refine architectures. However, these techniques often require significant computational resources and have struggled to deliver competitive performance compared to more recent advancements.

**Low-Cost NAS:** In response to the high computational demands of traditional NAS, low-cost methods have gained traction. Techniques like SynFlow Abdelfattah et al. (2021); Tanaka et al. (2020) and TSE Ru et al. (2021) estimate performance by analyzing gradient variations, while neural tangent kernel (NTK)-based methods such as TE-NAS Chen et al. (2021), KNAS Xu et al. (2021), and LGA Mok et al. (2022) leverage NTK assumptions to approximate performance. These approaches assume that the initial state of a model can predict its final performance Lee et al. (2019), but this assumption does not always hold in practice Mok et al. (2022); Zhang et al. (2024b). MOTE Zhang et al. (2024b) instead analyzes the loss landscape through linear interpolation during training. While these methods are computationally efficient, they rely on complex mathematical assumptions, limiting their generalizability across different search spaces Mok et al. (2022); Zhang et al. (2024b).

**Encoder-based NAS:** Unlike gradient-based or other low-cost NAS methods, encoder-based approaches Liu et al. (2018a); Luo et al. (2018); Wen et al. (2020); Dudziak et al. (2020); Wei et al. (2022); Wu et al. (2021) train on a small set of architecture–performance pairs to balance efficiency and accuracy. Early works Liu et al. (2018a); Luo et al. (2018); Wen et al. (2020); Dudziak et al. (2020) used GCNs to sample architecture batches, outperforming DARTS at lower cost. Weak-NAS Wu et al. (2021) showed that a simple MLP with evolutionary search could do even better, while BRP Dudziak et al. (2020) and RATs Zhang et al. (2023) enhanced encoders with specialized modules, and GATE/TA-GATE Ning et al. (2020; 2022) used simulated inputs for embedding extraction. Yet most still rely on graph-based encodings, where one-hot operation labels fail to represent unseen operations, forcing retraining for each search space. To address this, we introduce semantic features into architecture encoders.

**Contrastive LanguageImage Pre-training (CLIP):** CLIP Radford et al. (2021) is a vision-language foundation model that learns to align images and text in a shared embedding space. Trained on 400 million image–text pairs from the Internet, CLIP uses a contrastive learning objective to maximize the similarity between matching image–text pairs. At the intersection of NAS and vision-language model, GrowCLIP Deng et al. (2023) represents a data-driven automatic model growing algorithm inspired by CLIP. Despite the analogy between images and graphs, the potential of leveraging CLIP to facilitate NAS has not been considered in the open literature as far as we know.

## 3 CONTRASTIVE LEARNABLE UNIFYING ENCODER

The proposed Contrastive Learnable Unifying Encoder for NAS (CLUE-NAS) takes three types of input to represent a candidate architecture: bottom-up information flow, operation metrics, and top-down context features. The first two, similar to previous approaches Wen et al. (2020); Dudziak et al. (2020); Shen et al. (2021); Zhang et al. (2024a), treat the architecture as a graph, and the last one is derived using a CLIP's text encoder. CLUE-NAS aligns the information flow and operation metrics with context features, resulting in more robust embeddings. Subsequently, CLUE-NAS mimics human expert behaviors by qualitatively summarizing architecture performance through a coarse-

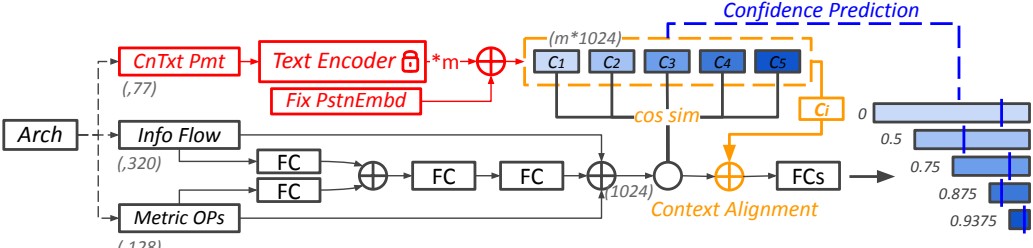

Figure 2: Overview of CLUE-NAS. Architecture prompts are encoded into context embeddings via CLIP's text encoder, while architectural topologies are transformed into graph-based embeddings. CLUE-NAS aligns these representations via cosine similarity and refines the alignment progressively for accurate prediction.

to-fine confidence prediction method. This coarse-to-fine strategy avoids overly rigid predictions and significantly enhances CLUE-NAS's ability to generalize to unseen architectures.

### 3.1 GRAPH-BASED EMBEDDING

The proposed CLUE-NAS framework aligns two distinct types of embeddings: a graph-based embedding that encodes the architectural topology, and a context embedding that captures high-level semantic cues to bridge the structural representation of neural architectures with more semantic features. Before introducing our core alignment mechanism, we first introduce the construction of the graph-based embedding. Abstractly, a candidate architecture $\mathcal{A}$ comprises various operations $O = \{O_1, O_2, \ldots, O_n\}$ (e.g., convolution, pooling, fully connected layers) and the corresponding information flow patterns, such as residual and dense connections. When operations are treated as nodes and the information flow as edges between them, representing an architecture as a graph becomes intuitive. In this formulation, the adjacency matrix encodes the edges, while the feature matrix represents the node attributes.

**Flattened Adjacency Matrix:** Our encoding of architectural edges is inspired by the insights of RATs Zhang et al. (2024a), which argue that a binary adjacency matrix is overly rigid. When coupled with Graph Convolutional Network (GCN) for feature extraction, such rigidity can have detrimental effects, often leading to worse performance than using a simple MLP. Following this observation, we flatten the adjacency matrix and zero-pad to a fixed length of 320 to form a vector representation, denoted as $\mathcal{A}_{flow}$. This vector is then directly processed by an MLP for feature extraction. The calculation process is shown in Fig. 2.

**Operation Metrics:** A common approach to encoding operations involves constructing a feature matrix using one-hot vectors. However, one-hot encoding serves as a hard label, which limits the model's ability to generalize to previously unseen architectures containing novel operations, as illustrated in Fig. 1. To overcome this limitation, we introduce a set of five soft metrics to describe each operation $O_i$, such as parameter count, FLOPs, latency, and other relevant attributes. These soft, proxy labels offer a more expressive and flexible representation of operations, thereby mitigating the drawbacks of one-hot encoding. The resulting feature matrix is flattened, zero-padded to length 128 (denoted $\mathcal{A}_{ops}$), and processed by an MLP. As shown in Tab. 4, this representation improves CLUE-NAS performance. Finally, as illustrated in Fig. 2, we concatenate the outputs of two MLPs applied to $\mathcal{A}flow$ and $\mathcal{A}_{ops}$, then feed the result into another MLP to obtain the graph-based embedding $G$, capturing the architecture's topology:

$$G = MLP_{\mathcal{A}}(MLP_f(\mathcal{A}_{flow}), MLP_o(\mathcal{A}_{ops})),$$ (1)

where $MLP_{\mathcal{A}}$, $MLP_f$, and $MLP_o$ are three multi-layer perceptrons responsible for transforming $\mathcal{A}$, $\mathcal{A}_{flow}$, and $\mathcal{A}_{ops}$ into their respective feature vectors.

### 3.2 ALIGN THE CONTEXT AND GRAPH-BASED EMBEDDINGS

To obtain a global contextual representation of an architecture $\mathcal{A}$, we construct a natural language prompt $\mathcal{A}pmt$ in the form: "The architecture has $N$ nodes: $O_1, O_2, ..., O_N$." This description is

encoded by CLIP's pre-trained text encoder $\mathcal{T}$ to produce a semantic embedding $\mathcal{T}(\mathcal{A}pmt)$. Leveraging CLIP's semantic abstraction, this embedding acts as a global prior that guides the learning of the graph-based representation—mirroring how human experts first form an intuitive overview before detailed structural analysis.

**Positional Embeddings for Confidence Score Division:** To emulate this coarse-to-fine reasoning process, the proposed CLUE-NAS framework introduces a contrastive alignment mechanism. Specifically, we embed the context representation with multiple levels of confidence, where each level reflects a progressively refined estimation of the candidate architecture's expected performance. Then, the graph-based embedding $G$ is aligned with these context representations, enabling it to absorb both high-level semantic priors and the associated confidence levels.

To distinguish between different levels of confidence without introducing additional learning complexity, we incorporate a set of fixed, non-learnable positional embeddings $E = \{E_i\}_{i=1,...,M}$, where $E_i = \frac{i}{M}$ and $M$ is a hyper-parameter used to divide the confidence level to $M$ slots. The effects of $M$ on performance improvement will be addressed in the experimental section. These scalar values are linearly spaced and serve to differentiate between context embeddings at different confidence levels. These levels later guide the CLUE-NAS in emulating the coarse-to-fine reasoning process commonly employed by human experts during architecture prediction. With $E_i$, a set of context embeddings $C$ can be constructed to represent varying degrees of confidence about the architecture's performance:

$$C = \{\mathcal{T}(\mathcal{A}_{pmt}) + E_i\}_{i=1,...,M}. \tag{2}$$

Each context embedding $C_i$ now jointly encodes both the semantic features of the architecture $\mathcal{A}$ and the corresponding confidence level. To further incorporate this confidence information into the learning objective, we assign a lower bound on the predicted performance for each confidence level. Specifically, for each $C_i$, we define the lower bound as $C_i^b = 1 - \frac{1}{2^{i-1}}, i = 1, 2, ...M$, such that higher confidence levels correspond to progressively higher lower bounds.

**Context Alignment:** During training, the ground-truth context embedding $C_{gt}$ is derived from test accuracy. The graph-based representation $G$ is then aligned with $C_{gt}$ via a joint loss $\mathcal{L}$ combining KL divergence and cosine similarity.

$$\mathcal{L}_{algn}(G, C_{gt}) = KL(G, C_{gt}) + (1 - \sigma(Sim(G, C_{gt}))), \tag{3}$$

where $Sim()$ is a similarity function and $\sigma$ is a sigmoid function. During inference, we compute the similarity between the graph-based embedding $G$ and each confidence-aware context embedding $C_i$ through an inner product. Then, we identify the context embedding $C_{\max}$ with the highest similarity:

$$C_{\max} = \arg \max_{C_i \in C} Sim(G, C_i). \tag{4}$$

Finally, we concatenate the graph-based embedding $G$ with its most semantically aligned context embedding $C_{\max}$ to form a robust and semantically enriched representation $[G, C_{max}]$.

**Confidence Prediction:** So far, CLUE-NAS has generated a semantically enriched embedding vector, denoted as $[G, C_{\max}]$. To emulate the initial coarse-grained assessment typically performed by human experts when evaluating a neural architecture, we introduce a predefined lower-bound estimate of performance, denoted as $C_{best}^b$. Notably, the value of $C_{best}^b$ is selected from a predefined set of candidates $C_1^b, C_2^b, \ldots, C_M^b$ based on the index of $C_{\max}$ (or best = the index of $C_{\max}$). This constraint enables CLUE-NAS to make informed predictions within a plausible performance range, rather than generating arbitrary outputs. Such a design mirrors the human decision-making process, where an initial rough estimation precedes a more precise evaluation. Specifically, the final target prediction $\mathcal{P}(\mathcal{A})$ is formulated as a bounded refinement over $C_{best}^b$:

$$\mathcal{P}(\mathcal{A}) = \mathcal{P}(G, C_{\max}) = C_{best}^b + \sigma(MLP([G, C_{\max}])) \times (1 - C_{best}^b), \tag{5}$$

where the embedding $[G, C_{\max}]$ is passed through a feedforward network MLP, and the output is scaled by a sigmoid function $\sigma$. To ensure it is proportional to the confidence gap, the normalized strength is scaled again by $(1 - C_{best}^b)$. We refer to this human-inspired coarse-to-fine refinement strategy as Confidence Prediction.

**Training Setting:** The term "architecture-accuracy pair" is defined as a candidate architecture and its corresponding test accuracy. Obtaining an architecture-accuracy pair is time-consuming; for instance, each pair can take thousands of GPU seconds on CIFAR-10 and even hundreds of thousands

Table 1: Spearman correlations for NASBench-101, NASBench-201, and NASBench-301 using encoders trained independently with three training budget across all three search spaces.

| Model | 100 pairs per NB | | | 50 pairs per NB | | | 5 pairs per NB | | |
|---|---|---|---|---|---|---|---|---|---|
| | NB101 | NB201 | NB301 | NB101 | NB201 | NB301 | NB101 | NB201 | NB301 |
| MLP | 0.427 | 0.664 | 0.535 | 0.205 | 0.589 | 0.246 | 0.161 | 0.177 | 0.061 |
| GCN | 0.362 | 0.416 | 0.489 | 0.122 | 0.428 | 0.183 | -0.318 | -0.269 | -0.122 |
| BiGCN | 0.603 | 0.386 | 0.482 | 0.468 | 0.649 | 0.393 | -0.166 | 0.095 | 0.056 |
| RATs-GCN | 0.623 | 0.653 | 0.420 | 0.447 | 0.633 | 0.339 | 0.170 | 0.359 | 0.181 |
| **CLUE-NAS** | **0.740** | **0.831** | **0.716** | **0.530** | **0.728** | **0.607** | **0.366** | **0.481** | **0.496** |

of GPU seconds on ImageNet. Therefore, minimizing the consumption of architecture-accuracy pairs throughout the search process is crucial for CLUE-NAS and other encoder-based NAS methods. To train the CLUE-NAS, we first sample a set of architectures from the search space and train them to obtain their corresponding accuracies, thereby forming architecture–accuracy pairs. The number of training pairs varies across experiments (5, 50, or 100), and the impact of this variation is discussed in the experimental section. All CLUE-NAS training runs share the same hyperparameter settings: the batch size is fixed at 16, and the training proceeds for a total of 200 epochs. The learning rate is initially set to 0.01 for the first 100 epochs and then reduced to 0.001 for the remaining 100 epochs. The overall training loss $\mathcal{L}_{Clue}$ comprises two components:

$$\mathcal{L}_{Clue} = \mathcal{L}_{BCE}(\mathcal{P}(\mathcal{A}), ACC_{gt}) + \mathcal{L}_{algn}(G, C_{gt}), \tag{6}$$

where $\mathcal{L}_{BCE}$ is a binary cross-entropy loss used for optimizing the confidence prediction mechanism, which is based on the confidence-bound estimation described earlier. The alignment loss $\mathcal{L}_{algn}$ previously introduced in Eq. (3), is designed to enforce consistency between architecture embeddings and contextual features.

## 4 EXPERIMENTAL RESULTS

We use three search spaces (all of them are cell-based search spaces): NASBench-101 consists of 423,621 candidates trained on CIFAR-10 for 108 epochs. NASBench-201 includes 15,625 candidates trained on CIFAR-10, CIFAR-100, and ImageNet-16-120 for 200 epochs each. NASBench-301 contains 57,189 candidates trained on CIFAR-10 for approximately 100 epochs.

### 4.1 COMPARISON CLUE-NAS AND ENCODERS IN NAS

**Independent Training Experiments:** We present a comparative analysis between CLUE-NAS and several baseline encoders that rely solely on topological representations. Experiments are conducted across three distinct NAS search spaces, where each model is independently trained under three data regimes: 100 training pairs per search space, 50 pairs per search space, and an extreme few-shot scenario with only 5 pairs per search space. As shown in Tab. 1, CLUE-NAS consistently achieves substantially higher Spearman correlation coefficients than competing methods across all training settings. While CLUE-NAS retains strong predictive performance even in the low training data regime, conventional baselines such as GCN and its variant BiGCN Wen et al. (2020) often exhibit negative correlations, indicating a complete collapse in predictive capacity. Although RATs-GCN Zhang et al. (2024a) is a self-attention-enhanced GCN variant, and it performs better than standard GCNs, it is still consistently outperformed by CLUE-NAS across all training data budgets. Under the 5 pairs setting, RATs-GCN achieves Spearman correlations of 17.0%, 35.9%, and 18.1% on NASBench-101, NASBench-201, and NASBench-301, respectively. In contrast, CLUE-NAS achieves 36.6%, 48.1%, and 49.6% on the same benchmarks, representing a substantial improvement. These results underscore the effectiveness and robustness of CLUE-NAS as an advanced encoder for neural architecture performance prediction.

**Joint Training Experiments:** We further present an additional set of experiments conducted on the same search spaces to evaluate the performance of the encoder after joint training. In this setup, we consider three different training budgets: 100, 50, and 5 architecture pairs per search space. These training pairs are then aggregated, resulting in a total of 300, 150, and 15 pairs, respectively, which are used to jointly train the model shared across all three search spaces in each pair's setting.

Table 2: Spearman correlations for NASBench-101, NASBench-201, and NASBench-301 using encoders trained jointly with three training budget across all three search spaces.

| Model | 100 pairs per NB | | | 50 pairs per NB | | | 5 pairs per NB | | |
|---|---|---|---|---|---|---|---|---|---|
| | NB101 | NB201 | NB301 | NB101 | NB201 | NB301 | NB101 | NB201 | NB301 |
| MLP | 0.527 | 0.643 | 0.598 | 0.241 | 0.580 | 0.368 | 0.292 | 0.214 | 0.074 |
| GCN | 0.315 | 0.360 | 0.308 | 0.125 | 0.176 | 0.174 | -0.036 | 0.147 | -0.146 |
| BiGCN | 0.635 | 0.658 | 0.569 | 0.400 | 0.529 | 0.386 | 0.108 | 0.467 | 0.206 |
| RATs-GCN | 0.583 | 0.634 | 0.473 | 0.296 | 0.560 | 0.355 | 0.083 | 0.459 | 0.320 |
| **CLUE-NAS** | **0.771** | **0.852** | **0.724** | **0.593** | **0.767** | **0.634** | **0.445** | **0.612** | **0.552** |

Table 3: Spearman correlations on three search spaces based on encoders that only train on one of them. Note that the training budget is 100, 50, and 5 per NASBench.

| Model | Dataset used for training | 100 pairs per NB | | 50 pairs per NB | | 5 pairs per NB | |
|---|---|---|---|---|---|---|---|
| | | NB201 | NB301 | NB201 | NB301 | NB201 | NB301 |
| MLP | NB101 | 0.263 | 0.056 | -0.108 | -0.019 | -0.121 | -0.111 |
| GCN | NB101 | -0.065 | -0.199 | -0.148 | -0.074 | -0.201 | -0.188 |
| BiGCN | NB101 | 0.446 | 0.383 | 0.427 | 0.221 | 0.210 | 0.022 |
| RATs-GCN | NB101 | 0.459 | 0.245 | 0.370 | 0.202 | -0.057 | -0.083 |
| **CLUE-NAS** | NB101 | **0.631** | **0.513** | **0.583** | **0.474** | **0.255** | **0.346** |
| | | NB101 | NB301 | NB101 | NB301 | NB101 | NB301 |
| MLP | NB201 | 0.363 | 0.129 | 0.315 | 0.019 | 0.081 | -0.228 |
| GCN | NB201 | -0.065 | -0.163 | -0.154 | 0.007 | -0.125 | -0.005 |
| BiGCN | NB201 | 0.332 | 0.235 | 0.131 | 0.121 | -0.243 | -0.364 |
| RATs-GCN | NB201 | 0.192 | 0.067 | 0.170 | 0.141 | 0.028 | -0.165 |
| **CLUE-NAS** | NB201 | **0.529** | **0.461** | **0.533** | **0.392** | **0.338** | **0.224** |
| | | NB101 | NB201 | NB101 | NB201 | NB101 | NB201 |
| MLP | NB301 | -0.003 | 0.054 | -0.005 | -0.052 | -0.130 | -0.041 |
| GCN | NB301 | 0.236 | 0.159 | -0.104 | -0.164 | -0.047 | -0.361 |
| BiGCN | NB301 | -0.074 | 0.024 | -0.120 | -0.036 | -0.111 | -0.025 |
| RATs-GCN | NB301 | -0.534 | -0.745 | -0.069 | -0.328 | -0.131 | -0.347 |
| **CLUE-NAS** | NB301 | **0.422** | **0.539** | **0.337** | **0.413** | **0.120** | **0.274** |

As shown in Tab. 2, the results demonstrate that CLUE-NAS exhibits a significant performance improvement. Compared to the results in Tab. 1, CLUE-NAS consistently benefits from joint training across all budget settings. Specifically, under the extreme low training pairs regime of only 5 pairs per search space, CLUE-NAS achieves impressive Spearman correlations of 44.5%, 61.2%, and 55.2% on NASBench-101, NASBench-201, and NASBench-301, respectively. These represent substantial improvements of 32.4%, 27.2%, and 11.3% over the independently trained CLUE-NAS.

In contrast, other methods, such as MLP, GCN, BiGCN, and RATs-GCN do not consistently benefit from joint training. In some cases, their performance even degrades. For example, GCN exhibits a drop in Spearman correlation under the 100 pairs setting, with decreases of -8.0%, -15.6%, and -58.7% on NASBench-101, NASBench-201, and NASBench-301, respectively. We attribute this to CLUE-NAS's ability to capture and integrate high-level semantic features of architectures, which enables it to extract transferable knowledge across different search spaces. In contrast, traditional NAS encoders lack this crucial capability, and thus, joint training may introduce noisy supervision, leading to unstable optimization and degraded performance.

**Unseen Architecture Experiments:** As shown in Tab. 2, CLUE-NAS benefits strongly from joint training, indicating its ability to capture higher-dimensional representations. To test this, we trained with 100, 50, and 5 pairs under disjoint search spaces—e.g., training on NASBench-101 and evaluating on NASBench-201/301 using Spearman correlation. This setup requires predicting performance on unseen architectures, a difficult task given differences across search spaces in layers, hyperparameters, and candidate operations (e.g., NASBench-301 adds dilated and separable convolutions absent from NASBench-101/201).

Tab. 3 shows the experimental results. CLUE-NAS consistently maintains a considerable level of predictive capability across different conditions. Notably, when trained on NASBench-101 with 100 pairs, CLUE-NAS achieves Spearman correlations of 63.1% and 51.3% on NASBench-201 and

Table 4: Ablation study of each component in CLUE-NAS.

| OP Metrics | Fixed Pstn Embd | Cntxt Align | Cnfd Pred | NB101 | NB201 | NB301 |
|:---:|:---:|:---:|:---:|:---:|:---:|:---:|
| | ✓ | ✓ | ✓ | 0.737 | 0.819 | 0.532 |
| ✓ | | ✓ | ✓ | 0.722 | 0.763 | 0.595 |
| ✓ | ✓ | | ✓ | 0.726 | 0.802 | 0.619 |
| ✓ | ✓ | ✓ | | 0.693 | 0.796 | 0.665 |
| ✓ | ✓ | ✓ | ✓ | **0.771** | **0.852** | **0.724** |

Table 5: Ablation study of hyper-parameter $M$ for context embeddings

| $M$ | 100 pairs per NB | | | 50 pairs per NB | | | 5 pairs per NB | | |
|:---:|:---:|:---:|:---:|:---:|:---:|:---:|:---:|:---:|:---:|
| | NB101 | NB201 | NB301 | NB101 | NB201 | NB301 | NB101 | NB201 | NB301 |
| 3 | 0.713 | 0.805 | 0.641 | 0.576 | 0.749 | 0.523 | 0.363 | 0.503 | 0.299 |
| **5** | **0.771** | **0.852** | **0.724** | **0.593** | **0.767** | **0.634** | **0.445** | **0.612** | **0.552** |
| 7 | 0.732 | 0.824 | 0.651 | 0.580 | 0.765 | 0.579 | 0.382 | 0.527 | 0.324 |

NASBench-301, respectively, despite these architectures being entirely unseen during training. In contrast, BiGCN and the more advanced RATs-GCN perform significantly worse under the same conditions, with GCN even exhibiting negative correlations, indicating a complete loss of predictive ability. Furthermore, under the extreme condition of only 5 training pairs, CLUE-NAS still retains predictive ability. Although its performance is reduced, all baseline methods (MLP, GCN, BiGCN, and RATs-GCN) entirely fail under this setting.

These results further demonstrate CLUE-NAS's capacity to capture high-level semantic representations of candidate architectures. As a result, it can generalize across different search spaces and operate effectively under few-shot scenarios, and these capabilities are lacking in prior encoders. Fig. 4 shows more visualization results, please refer to Appendix B.

## 4.2 Ablation study of CLUE-NAS

**Impact of Key Components:** CLUE-NAS comprises four key components: (1) replacing one-hot encodings with Operation Metrics (OP Metrics), (2) using context embeddings as fixed positional encodings (Fixed Pstn Embd), (3) aligning context and graph embeddings (Cntxt Algn), and (4) coarse-to-fine prediction (Cnfd Pred). We conduct ablations under the 100-pair joint training setting (Tab. 4), finding that removing any component degrades performance—highlighting the critical role of each design choice.

**Impact of Context Embedding Length** CLUE-NAS introduces a hyperparameter $M$ that determines the length of the context embeddings. This length controls the granularity of the lower bound distribution of confidence levels $C_i^b$. In essence, a larger $M$ results in a finer division of the confidence spectrum, allowing the confidence prediction module to include more branches that specifically target high-performing architectures.

We conducted an ablation study (Tab. 5) on different values of $M$, finding the best performance at $M = 5$. With $M = 3$, context embeddings are too short, yielding coarse representations that weaken the coarse-to-fine prediction. At $M = 7$, confidence bins become extreme (e.g., $C_7^b = 0.984375$), leaving few training pairs per bin and introducing redundancy and noise. Thus, setting $M$ too low or too high degrades CLUE-NAS performance.

## 4.3 Comparison between Finetune-free CLUE-NAS and Other NASs

As shown in Fig.1, CLUE-NAS demonstrates strong generalization, enabling deployment without finetuning. To evaluate its performance, we assess it on the three sub-datasets of NASBench-201, using only 100 training pairs from NASBench-101 and 100 from NASBench-301, without including any data from NASBench-201. As a result, CLUE-NAS operates on NASBench-201 with zero adaptation, and the search cost is determined solely by evaluation. We further combine CLUE-NAS with an evolutionary algorithm similar to that in Zhang et al. (2024b)(for a comparison of sampling methods, please refer to Appendix C.), referred to as CLUE-NAS-F. In addition, we propose

Table 6: Comparison of the proposed CLUE-NAS and other NAS methods on NASBench-201. We have selected the most recent and relevant benchmark methods for each category. Note that 'Cost (s)' means the total cost in GPU seconds, containing the training and evaluation cost. The best-performing result within each category is highlighted in bold.

| Type | Model | CIFAR-10 | | CIFAR-100 | | ImgNet-16 | |
|---|---|---|---|---|---|---|---|
| | | Acc(%) | Cost(s) | Acc(%) | Cost(s) | Acc(%) | Cost(s) |
| Low-cost | KNAS (k=20) Xu et al. (2021) | 93.38 | 4.4K | 70.78 | 9.2K | 44.63 | 20K |
| | Eigen-NAS (k=20) Zhu et al. (2022) | 93.46 | 4.4K | 71.42 | 9.2K | 45.53 | 20K |
| | LGA Mok et al. (2022) | 94.30 | 3.6K | 72.42 | 5.4K | 45.30 | 3.6K |
| | MOTE-NAS (k=10) Zhang et al. (2024b) | 94.15 | 4.2K | 72.54 | 4.3K | 46.38 | 11.3K |
| LLM | GENIUS Zheng et al. (2023) | 93.79 | 8.0M | 70.91 | 8.0M | 44.96 | 25.0M |
| | LLMatic Nasir et al. (2024) | 94.26 | 8.0M | 71.62 | 8.0M | 45.87 | 25.0M |
| Encoder | Neural Predictor Wen et al. (2020) | 94.07 | 840.0K | 72.18 | 840.0K | 46.39 | 2.4M |
| | WeakNAS Wu et al. (2021) | 94.23 | 840.0K | 73.42 | 840.0K | 46.79 | 2.4M |
| | Proxy-BO Shen et al. (2021) | - | - | 73.48 | 1.2M | 47.18 | 3.2M |
| | Arch-Graph Huang et al. (2022) | - | - | 73.38 | 840.0K | - | - |
| | RATs-NAS Zhang et al. (2023) | 93.98 | 14.8K | 72.35 | 13.8k | 45.39 | 34.7k |
| | **CLUE-NAS-F** | **94.34** | 451.2K | **73.51** | 426.7K | **47.31** | 1.3M |
| | **CLUE-NAS-EF** | 94.30 | 12.8K | 72.64 | 12.4K | 46.79 | 31.8K |

CLUE-NAS-EF, which accelerates the search using early-stopping test accuracy, with a trade-off in performance. Results are presented in Tab. 6, and the analysis is structured in three parts: comparison with encoder-based, LLM-based, and low-cost NAS methods.

**Encoder-based NAS:** Compared to encoder-based NAS, CLUE-NAS-F finds superior architectures. Even against RATs-NAS Zhang et al. (2023), CLUE-NAS-EF achieves better results at smaller search costs. Unlike other predictors that require finetue on specific search spaces, both CLUE-NAS-F and CLUE-NAS-EF operate in a finetue-free manner, highlighting advantage of CLUE-NAS in practical applications.

**LLM-based NAS:** In recent years, several attempts have been made to leverage large language models (LLMs) for NAS, as exemplified by GENIUS Zheng et al. (2023) and LLMatic Nasir et al. (2024). These methods are expected to offer strong semantic representations. However, they remain in an early exploratory stage and fail to integrate the topological features that are intrinsic to traditional encoder-based NAS approaches. As a result, their performance is suboptimal. In contrast, CLUE-NAS-F outperforms both GENIUS and LLMatic, achieving superior accuracies of 94.34%, 73.51%, and 47.30%, while also demonstrating significantly lower search costs compared to LLMatic. The more efficient CLUE-NAS-EF variant achieves 94.30%, 72.64%, and 46.79%, still surpassing both baselines while further improving search efficiency. Note that the reported search costs for GENIUS and LLMatic are estimated based on Nasir et al. (2024), which involves 2000 architecture-accuracy pairs during training and evaluation stages.

**Low-cost NAS:** CLUE-NAS combines an encoder with semantic priors, diverging from speed-focused low-cost NAS methods. Though more computationally intensive, it delivers higher accuracy: CLUE-NAS-ES achieves 46.79 on ImageNet-16 in 31.8K GPU seconds, surpassing MOTE-NAS (46.38%), while CLUE-NAS-F reaches 47.30%, near the NAS-Bench-201 upper bound. Thus, CLUE-NAS is ideal when top-tier performance is required.

## 5 CONCLUSIONS

This paper presents CLUE-NAS, a novel encoder for Neural Architecture Search (NAS) that integrates both topological representations and high-level semantic features. In contrast to prior encoders that primarily focus on architectural topology, CLUE-NAS captures deeper semantic cues, enhancing search efficiency and improving generalization to unseen architectures. It employs a coarse-to-fine prediction strategy: first estimating a lower bound of model performance, then refining the prediction for greater stability. Experiments on NASBench-101/201/301 show that CLUE-NAS consistently outperforms traditional encoders and matches state-of-the-art NAS methods, while requiring as few as five training pairs. Though slower than low-cost NAS approaches, it enhances the practicality of encoder-based NAS and highlights the value of semantic priors from language models, offering a promising path toward more efficient and interpretable NAS.

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

# A  APPENDIX

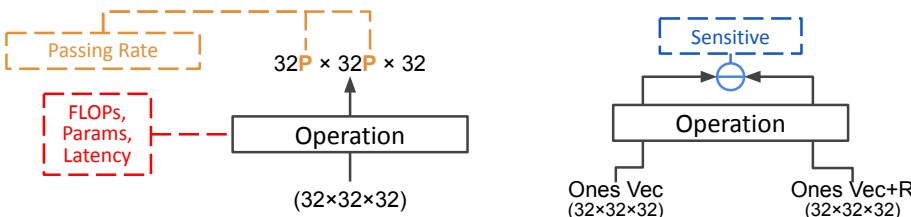

Figure 3: This figure illustrates the computation methods for different metrics. The left half of the figure presents the calculation of FLOPs, parameters, latency, and passing rate, while the right half demonstrates the computation of numerical sensitivity.

# B  IMPLEMENTATION DETAIL OF METRIC OPERATIONS

As previously mentioned, metric embedding utilizes five($c = 5$) distinct metrics to characterize the operations of a candidate architecture: [Parameters, FLOPs, Latency, Passing Rate, and Numerical Sensitivity], and $\phi_1$ to $\phi_5$ represent these five metrics. For a chosen operation, such as convolution or depthwise convolution, these metrics can only be computed once the operation is instantiated within a concrete model. To achieve this, we construct a single layer model centered on the chosen operation, with an input shape of (32,32,32), as illustrated in Fig. 3. Then every operation is described using this simple single-layer model and the five aforementioned metrics. In the following, we detail the computation of these metrics.

## B.1  FLOPS, PARAMETERS, AND LATENCY.

Once the single-layer model is constructed, obtaining these three metrics becomes straightforward. FLOPs represent the total number of multiplication operations required by the operation, while the parameters indicate the number of learnable weights within the operation. These two metrics reflect the theoretical computational cost in terms of both time and memory. In contrast, latency measures the actual time required for a single forward pass on hardware, providing a direct indication of the real-world execution cost. latency is calculated on the AMD Ryzen 7 PRO 5875U. We believe that these three metrics already capture essential properties of various operations.

## B.2  PASSING RATE.

This metric quantifies the proportion of features preserved or, conversely, lost during processing. In simpler terms, it represents the downsampling ratio. It is well-known that downsampling refines the original features, but it can also discard crucial information. To ensure a more comprehensive characterization of operations, we explicitly incorporate the downsampling ratio into our evaluation. As shown in the left part of Fig. 3, we compute the difference in the ratio between the output (height, width) and the input to obtain this metric.

## B.3  NUMERICAL SENSITIVITY.

This metric measures how sensitive an operation is to changes in input values. As shown in the right part of Fig. 3, we simultaneously input two sets of data into the operation. The first input is a (32,32,32) matrix filled with ones, while the second is the same matrix but with element-wise additions of randomly sampled values between 0 and 1. We then compute the difference between the outputs of these two inputs. This process is repeated 1000 times, and we take the average difference as the final measurement. This approach quantifies the operation's sensitivity to numerical variations. For example, due to their different downsampling mechanisms, max pooling and average pooling exhibit different levels of sensitivity to input perturbations.

# C  VISUALIZATIONS OF ENCODERS

We present t-SNE visualizations in Fig. 1 (partial results) and Fig. 4 (complete results). Each encoder is trained with only 100 architecture pairs sampled from one of the NASBench-101, NASBench-

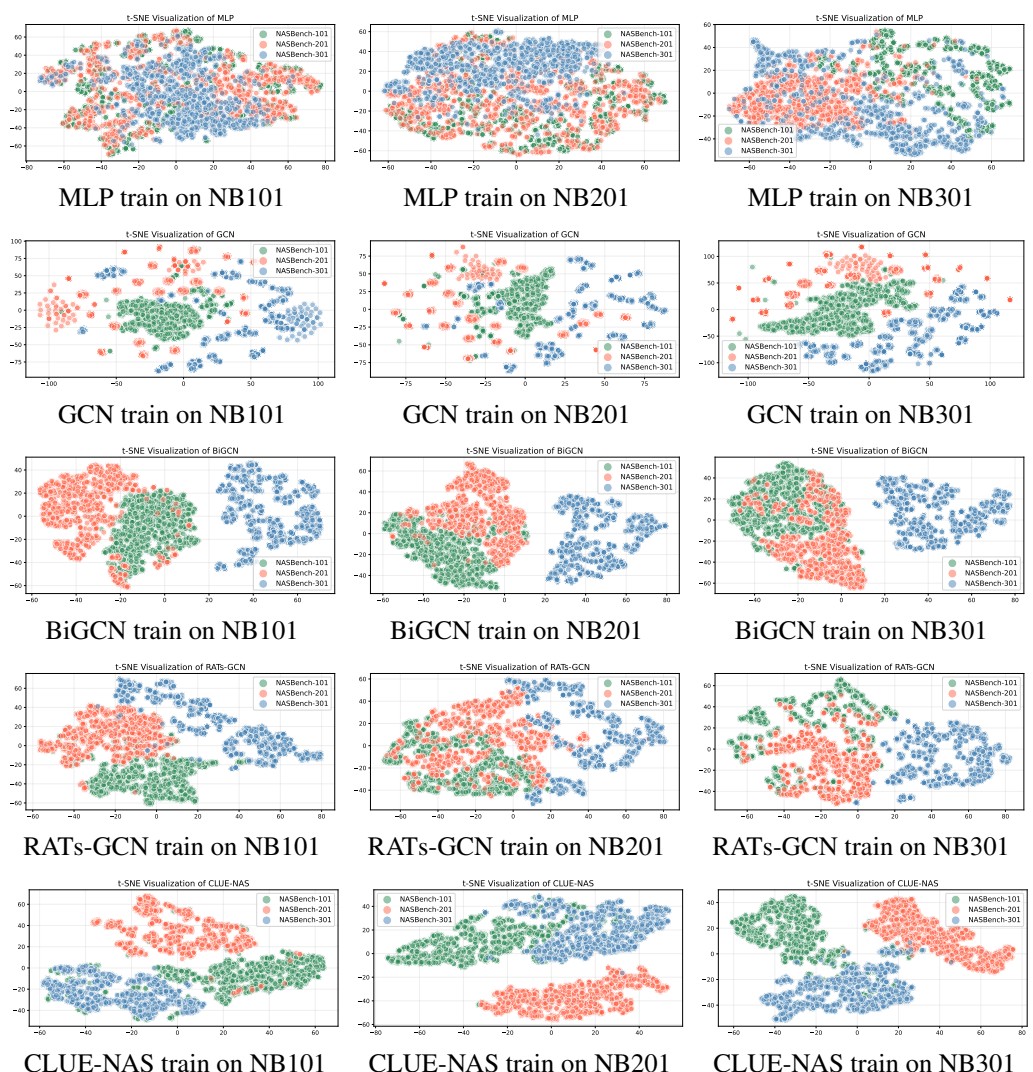

Figure 4: t-SNE visualizations of various encoders.

201, or NASBench-301 search spaces. After training, we randomly sample 1,000 architectures from each of the three search spaces and extract their embeddings using the trained encoders. These embeddings are then visualized using t-SNE. In the visualizations, different colors represent different search spaces. Note that the x- and y-axis values have no intrinsic meaning due to the nature of the t-SNE algorithm; the focus is instead on the spatial distribution of the embeddings learned by the encoders.

From the results, we observe that CLUE-NAS is able to clearly distinguish architectures from different, unseen search spaces, indicating that its learned representations capture higher-level semantic information. In contrast, the other encoder-based methods fail to achieve such separation, highlighting CLUE-NAS's superior ability to generalize across diverse architecture distributions, that is an ability that traditional encoder-based NAS methods lack.

## D    CLUE-NAS WITH SAMPLING STRATEGIES

Current predictors are often paired with sampling strategies to enhance search efficiency. We evaluated the synergy between CLUE-NAS and two widely used sampling strategies: Random (Rand) sampling and Evolutionary (Evo) sampling. Tab. 7 presents the performance of various predictors under these strategies. Here, the budget represents the total number of architecture-accuracy

Table 7: Performance comparisons between the top-30 accuracy (for Rand) and top-1 accuracy (for Evo) of architectures found on three search spaces using CLUE-NAS and other predictors.

| Model | Strategy | Budget | NB101 | NB201 | NB301 |
|---|---|---|---|---|---|
| MLP | rand | 200+30 | 0.9362 | 0.9395 | 0.9442 |
| GCN | rand | 200+30 | 0.9358 | 0.9404 | 0.9441 |
| BiGCN | rand | 200+30 | 0.9361 | 0.9407 | 0.9439 |
| RATs-GCN | rand | 200+30 | 0.9373 | 0.9398 | 0.9447 |
| **CLUE-NAS** | rand | **100+30** | **0.9392** | **0.9418** | **0.9462** |
| **CLUE-NAS** | rand | 200+30 | **0.9401** | **0.9437** | **0.9470** |
| MLP | evo | 200 | 0.9371 | 0.9410 | 0.9447 |
| GCN | evo | 200 | 0.9368 | 0.9411 | 0.9449 |
| BiGCN | evo | 200 | 0.9372 | 0.9428 | 0.9450 |
| RATs-GCN | evo | 200 | 0.9380 | 0.9429 | 0.9450 |
| **CLUE-NAS** | evo | **60** | **0.9390** | **0.9437** | **0.9469** |
| **CLUE-NAS** | evo | **90** | **0.9398** | **0.9437** | **0.9476** |

pairs used throughout the search process, including both predictor training and evaluation (note: evolutionary sampling does not require an evaluation stage). NB101, NB201, and NB301 denote the accuracy of architectures discovered by the predictors on NASBench-101, NASBench-201, and NASBench-301, respectively. The results demonstrate that CLUE-NAS consistently outperforms other predictors in discovering high-accuracy architectures. With random sampling, CLUE-NAS achieves outstanding performance with a budget of just 130 (100 for training and 30 for evaluation), surpassing predictors that use a budget of 230 (200 for training and 30 for evaluation). Similarly, under evolutionary sampling, CLUE-NAS delivers exceptional results with a budget of only 60, significantly outperforming other predictors that require a budget of 200—regardless of the search space. These findings highlight CLUE-NAS's strong adaptability to different sampling strategies, proving that it does not depend on any specific strategy to achieve superior results. Its performance remains consistently exceptional across diverse conditions.

# E   DISENTANGLING SEMANTIC UNDERSTANDING FROM BENCHMARK STYLE RECOGNITION

To examine whether the model's generalization originates from semantic understanding or from benchmark-specific style recognition, we conducted an experiment in which the operation names in NAS-Bench-101/201/301 were anonymized. Concretely, identifiers such as `nor_conv_3x3` or `skip_connect` were replaced with placeholders like `op1`, `op2`, and `op3` during evaluation, while the training procedure was kept unchanged. The results, based on 100 training pairs, are summarized in Table 8. We consistently observed a decline in performance across all cases when operator names were anonymized. This suggests that the graph representation, which is constructed from quantitative descriptors such as FLOPs, parameters, and latency, continues to provide a reliable source of information independent of the textual labels. At the same time, the drop in accuracy indicates that the model also benefits from the semantic cues carried by meaningful operation names. The CLIP text encoder, pretrained on large-scale web data, can extract contextual information from descriptors such as `conv` or `sepconv`, but this advantage disappears once the names are replaced by arbitrary placeholders. These findings highlight the dual role of structural metrics and semantic operation descriptors in supporting the model's generalization ability.

Table 8: Cross-dataset evaluation results with anonymized operator names.

| | **Eval on NS101** | **Eval on NS201** | **Eval on NS301** |
|---|---|---|---|
| *Train on NS101* | 76.56% | 69.89% | 43.34% |
| *Train on NS201* | 52.74% | 82.78% | 36.99% |
| *Train on NS301* | 34.29% | 37.40% | 70.82% |

## F    EFFECTIVENESS OF THE COARSE-TO-FINE STRATEGY

To assess the coarse-to-fine strategy, we evaluated context alignment and confidence prediction under independent training, where each benchmark was trained and tested separately. As shown in Table 9, performance is slightly lower than in joint training, but removing either component further degrades accuracy, with the largest drop from removing both. This confirms that both modules are beneficial and that the coarse-to-fine strategy remains effective even within a single benchmark.

Table 9: Ablation study on the impact of context alignment and confidence prediction under independent training.

|  | Eval on NS101 | Eval on NS201 | Eval on NS301 |
|---|---|---|---|
| – | 73.60% | 82.63% | 71.23% |
| w/o Context Align | 70.92% | 78.56% | 68.89% |
| w/o Cnfd Pred | 70.80% | 76.67% | 66.18% |
| w/o Context Align, Cnfd Pred | 65.95% | 77.49% | 54.41% |

## G    EXPLAINABILITY ANALYSIS VIA SHAP

To probe interpretability, we applied SHAP to the final MLP predictor to measure the relative importance of graph- and context-derived features. As shown in Table 10, context embeddings often dominate, but graph embeddings consistently retain significant weight. This indicates that CLUE-NAS combines semantic cues and structural metrics, preserving architectural reasoning rather than collapsing into benchmark style recognition.

Table 10: SHAP-based feature attribution analysis of graph embedding vs. context embedding.

|  | Eval on NS101 | | Eval on NS201 | | Eval on NS301 | |
|---|---|---|---|---|---|---|
|  | Graph SHAP | Context SHAP | Graph SHAP | Context SHAP | Graph SHAP | Context SHAP |
| *Train on NS101* | 0.2495 | 0.5526 | 0.2441 | 0.6176 | 0.1493 | 0.2619 |
| *Train on NS201* | 0.4468 | 0.7047 | 0.3516 | 0.6821 | 0.2389 | 0.5241 |
| *Train on NS301* | 0.1870 | 0.2036 | 0.2147 | 0.1971 | 0.1632 | 0.2395 |

## H    ANALYSIS ON THE DISTRIBUTION OF ACCURACY AND CONFIDENCE BOUNDS

To clarify the role of the coarse prediction mechanism, we conducted additional analyses regarding the distribution of the ground-truth accuracy $ACC_{gt}$, the proposed lower bounds $C^b$, and the selected bounds $C^b_{\text{best}}$.

### H.1    COMPARISON OF THE DISTRIBUTIONS OF $ACC_{gt}$ AND UNIFORM LOWER BOUNDS

We first sampled 1000 architectures each from NAS-Bench-101/201/301 and examined the distribution of their $ACC_{gt}$ values. As shown in Table 11, most samples are concentrated in the interval $[0.8, 1.0]$. This skewed distribution indicates that training directly with uniform bounds would lead to a severe imbalance problem, making such a design infeasible.

Table 11: Distribution of $ACC_{gt}$ values across NAS-Bench datasets.

|  | [0.0,0.2] | [0.2,0.4] | [0.4,0.6] | [0.6,0.8] | [0.8,1.0] |
|---|---|---|---|---|---|
| *Distribution of NS101* | 1 | 6 | 3 | 13 | 977 |
| *Distribution of NS201* | 19 | 0 | 6 | 68 | 907 |
| *Distribution of NS301* | 0 | 0 | 0 | 3 | 997 |

## H.2 Distribution based on proposed lower bounds $C^b$

We further considered the distribution based on the proposed lower bounds $C^b$, as shown in Table 12. Here, the upper bound is fixed at 1.0 (the theoretical maximum for accuracy), while overlapping ranges are introduced to alleviate label imbalance. This design not only differentiates confidence levels but also stabilizes the coarse prediction mechanism, as confirmed in the ablation study.

Table 12: Distribution of architectures under the proposed lower bounds $C^b$.

|                        | [0.1,1.0] | [0.5,1.0] | [0.75,1.0] | [0.875,1.0] | [0.9375,1.0] |
|------------------------|-----------|-----------|------------|-------------|--------------|
| *Distribution of NS101* | 1000      | 995       | 989        | 884         | 2            |
| *Distribution of NS201* | 1000      | 969       | 944        | 764         | 7            |
| *Distribution of NS301* | 1000      | 1000      | 1000       | 994         | 208          |

## H.3 Comparison between $C_{gt}^b$ and selected $C_{\text{BEST}}^b$

To further investigate how the model utilizes the proposed lower bounds during inference, we trained CLUE-NAS with 100 training pairs and sampled 3000 architectures (1000 from each NASBench). Table 13 shows the distribution of $C_{\text{best}}^b$ selected during prediction, compared with the distribution of ground-truth lower bounds $C_{gt}^b$.

Table 13: Comparison between the distribution of ground-truth lower bounds $C_{gt}^b$ and selected bounds $C_{\text{best}}^b$ during prediction.

|                                  | [0.1,1.0] | [0.5,1.0] | [0.75,1.0] | [0.875,1.0] | [0.9375,1.0] |
|----------------------------------|-----------|-----------|------------|-------------|--------------|
| *Distribution of $C_{best}^b$*   | 56        | 39        | 217        | 2600        | 88           |
| *Distribution of $C_{gt}^b$*     | 33        | 49        | 289        | 2387        | 242          |

Table 14: Spearman correlation across ten independent runs with five randomly sampled architectures. Results are reported for CLUE-NAS and the MLP baseline on three benchmarks (NS101, NS201, NS301).

|                  | 1st     | 2nd    | 3rd    | 4th    | 5th    | 6th     | 7th     | 8th    | 9th    | 10th   | Mean   |
|------------------|---------|--------|--------|--------|--------|---------|---------|--------|--------|--------|--------|
| *NS101 (CLUE-NAS)* | 55.26% | 36.75% | 33.24% | 53.67% | 47.52% | 18.53%  | 9.56%   | 42.32% | 47.14% | 54.46% | 39.85% |
| *NS201 (CLUE-NAS)* | 37.74% | 63.42% | 49.38% | 43.87% | 46.83% | 31.98%  | 66.86%  | 27.26% | 25.98% | 68.43% | 46.18% |
| *NS301 (CLUE-NAS)* | 38.85% | 38.67% | 56.43% | 61.76% | 68.74% | 55.25%  | 33.04%  | 45.80% | 57.11% | 59.45% | 51.53% |
| *NS101 (MLP)*      | 30.36% | -3.61% | 41.52% | 44.23% | 11.26% | 8.50%   | 13.50%  | 46.39% | 3.92%  | 10.83% | 20.69% |
| *NS201 (MLP)*      | 11.97% | 1.41%  | 31.49% | 38.46% | 6.43%  | -4.26%  | 5.60%   | 44.98% | -6.65% | -5.85% | 12.36% |
| *NS301 (MLP)*      | -19.38% | -2.27% | 6.22%  | 7.51%  | 18.40% | -17.68% | -11.94% | 23.94% | 10.07% | 22.06% | 3.69%  |

The distributions of $C_{gt}^b$ and $C_{\text{best}}^b$ are closely aligned, indicating that CLUE-NAS benefits from the design of the proposed lower bounds $C^b$ and is able to make effective coarse judgments of candidate architectures during prediction.

## I Robustness Analysis with Multiple Runs of Five-Architecture Subsets

To test robustness under an extreme five-pair setting, we ran ten trials on NAS-Bench-101/201/301, each with randomly sampled pairs. CLUE-NAS was trained and evaluated independently, with results summarized in Table 14 alongside an MLP baseline. CLUE-NAS shows substantially higher robustness, maintaining reasonable correlations, whereas the MLP baseline is unstable and often negative—underscoring the difficulty of this regime and the resilience of CLUE-NAS.

## J Comparison under Various Budgets

We evaluated CLUE-NAS, RATS-NAS (encoder/predictor baseline), and MOTE-NAS (low-cost baseline) on NASBench-101/201/301 under a 100-pair budget. For CLUE-NAS and RATS-NAS,

70 pairs were used for training, with the top 30 predictions validated; MOTE-NAS directly selected the top 100 predictions without training. Each experiment was repeated 10 times with different samples, and averaged results are reported in Table 15.

Table 15: Best architecture accuracy under a training budget of 100 pairs.

|  | NASBench-101 | NASBench-201 | NASBench-301 |
|---|---|---|---|
| MOTE-NAS | 93.75% | 94.13% | 94.43% |
| RATS-NAS | 93.67% | 93.92% | 94.49% |
| CLUE-NAS | 93.82% | 94.22% | 94.60% |

Table 16: Best architecture accuracy of MOTE-NAS under a budget of 10 pairs.

|  | NASBench-101 | NASBench-201 | NASBench-301 |
|---|---|---|---|
| MOTE-NAS | 93.72% | 93.91% | 94.25% |

Table 17: Best architecture accuracy of RATS-NAS and CLUE-NAS under a budget of 10 pairs.

|  | NASBench-101 | NASBench-201 | NASBench-301 |
|---|---|---|---|
| RATS-NAS | 91.57% | 92.46% | 93.41% |
| CLUE-NAS | 93.17% | 93.43% | 94.17% |

To further evaluate performance under extremely limited budgets, we decreased the number of pairs to only 10. In this setting, CLUE-NAS and RATS-NAS used 7 pairs for training and validated on their top-3 predicted architectures, while MOTE-NAS directly validated the top-10 predicted architectures. Results are reported in Table 16 and Table 17. Results show that CLUE-NAS consistently finds strong architectures under a 100-pair budget. With only 10 pairs, however, encoder-based methods (RATS-NAS and CLUE-NAS) degrade sharply, while MOTE-NAS remains stable. This highlights that low-cost NAS scales better under tight supervision, whereas encoder-based frameworks gain more from larger budgets.

