# OpenReview forum: "CLUE-NAS: A CLIP-Inspired Contrastive Learnable Unifying Encoder for Neural Architecture Search"
_ICLR.cc/2026/Conference — Submitted to ICLR 2026_

### Official Review · Reviewer_s1Ro · 2025-10-23

**Soundness:** 2
**Presentation:** 2
**Contribution:** 1
**Rating:** 2
**Confidence:** 5

**Summary:**

This paper proposes a contrastive learning-based neural performance predictor for Neural Architecture Search (NAS). Specifically, the proposed technique, Contrastive Learnable Unifying Encoder (CLUE), features a CLIP encoding of the architecture in order to enable cross-search space prediction. The method is evaluated on three popular NAS-Benchmarks and ablation studies are provided.

**Strengths:**

- The paper makes use of CLIP encodings for NAS.
- The figures are decently well formatted.
- The paper does not simply predict performance but considers prediction in a confidence interval, which is interesting.

**Weaknesses:**

- This paper is not well-motivated and much of the reasoning in the introduction is incorrect. For instance, the paragraph beginning on line 73 about how human's evaluate architectures - according to their end-to-end metrics like accuracy, FLOPs, etc., not cryptic topologies.
- The novelty of this work is limited. Generalized encodings for cell-based NAS was achieved by CDP [1] in NeurIPS 2022. CL was used for NAS before that [2].
- Evaluation on NAS-Bench-{101, 201, 301} for just CIFAR-10 has not been sufficient for a while now. Consider another benchmark like TransNASBench [3] that explores other, more applicable tasks.
- Experimental results are not that impressive, e.g., Table 3.
- Paper formatting is decent but there are glaring formatting issues and the floats are not well put together at all. Fig. 2 is hard to read, text of Fig. 1 requires a magnifying glass to read.

References:

[1] https://proceedings.neurips.cc/paper_files/paper/2022/hash/572aaddf9ff774f7c1cf3d0c81c7185b-Abstract-Conference.html

[2] https://arxiv.org/abs/2103.05471

[3] https://arxiv.org/abs/2105.11871

**Questions:**

N/A; see detailed weaknesses.

---

### Official Review · Reviewer_98k1 · 2025-10-25

**Soundness:** 2
**Presentation:** 3
**Contribution:** 2
**Rating:** 2
**Confidence:** 5

**Summary:**

This paper introduces CLUE-NAS, a NAS method which enhances encoder-based performance prediction by integrating both structural and semantic representations of architectures. Unlike traditional graph-based encoders, CLUE-NAS uses CLIP's text encoder to generate semantic embeddings from natural language descriptions of architectures and aligns them with graph-based embeddings via contrastive learning. This approach enables better generalization to unseen operations and search spaces, improves performance in low-data regimes, and achieves competitive results on NASBench-101, NASBench-201, and NASBench-301 without requiring fine-tuning.

**Strengths:**

+ The t-SNE visualizations for different encoders are nicely done.

+ The idea is clearly presented and easy to understand.

**Weaknesses:**

+ The experimental validation is limited. Specifically, the experimental validations are limited to existing NAS benchmarks, such as NAS-Bench-101, NAS-Bench-201, and NAS-Bench-301. It is unclear whether the proposed method can be adopted on larger-scale benchmarks such as ImageNet-1K. If not, the practicality of CLUE-NAS is quite limited.

+ The motivation for choosing CLIP is not really convincing to me. Because there are multiple multi-modal models which are stronger than CLIP, e.g., Qwen2.5-VL and InternVL, why not choose these models to achieve the better performance?

+ There comparison regarding the search cost seems to be unfair. In particular, the encoder needs to be trained based on architecture-performance pairs in existing NAS benchmarks. I believe that the training time for such architecture-performance pairs needs to be added to the overall search cost, to ensure the fair comparison.

+ The scalability of CLUE-NAS is not fully evaluated. I wonder if CLUE-NAS is still effective on other kind of tasks beyond image classification. Additional experiments on NAS-Bench-NLP or TransNAS-Bench-101 would be helpful.

+ The format of the references needs to be modified. Please use \citep{} instead of \cite{}.

**Questions:**

Please refer to the Weaknesses part.

---

### Official Review · Reviewer_MFSo · 2025-10-31

**Soundness:** 3
**Presentation:** 3
**Contribution:** 3
**Rating:** 4
**Confidence:** 4

**Summary:**

The authors of this paper introduce CLUE-NAS, an encoder-based Neural Architecture Search (NAS) framework that combines graph-based structural representations of neural networks with semantic context embeddings derived from CLIP’s text encoder. The goal is to improve the performance and interpretability of NAS. The experimental results show that CLUE-NAS outperforms prior NAS encoders (MLP, GCN, RATs-GCN) and competes with state-of-the-art NAS baselines.

**Strengths:**

- The idea of incorporating semantic priors derived from a large-scale vision–language model like CLIP into NAS is novel. It addresses a fundamental limitation of previous NAS encoders, specifically their inability to interpret architectures across heterogeneous search spaces or to generalize beyond the specific operation set used during training. CLUE-NAS leverages language-based descriptions to introduce a more expressive representation for the architectural operations that allows the model to reason about unseen operations.

-CLUE-NAS matches or outperforms several recent NAS approaches, including low-cost and LLM-based NAS variants, while requiring fewer architecture–accuracy pairs and achieving competitive computational efficiency.

**Weaknesses:**

- While CLUE-NAS employs CLIP’s text encoder to produce global semantic embeddings representing the entire architecture, the authors do not examine alternative integration strategies. An important baseline would involve using CLIP embeddings as node-level features. Such a setup would test whether the main benefit of CLIP arises from its global semantic alignment or whether its features can enhance structural reasoning at the node level without any contrastive objective.

- The paper omits a discussion of prior research that explored operation embedding techniques as an alternative to one-hot operation encoding in NAS, such as [1]. Earlier works have shown that learning continuous embeddings for operations can improve the performance. While CLUE-NAS introduces semantics through a distinct mechanism, using “operation metrics” such as FLOPs, latency, and parameter counts combined with contrastive language alignment, this idea conceptually parallels those efforts to move beyond discrete operation representations. Acknowledging this line of work would help clarify how CLUE-NAS builds upon and differs from previous approaches, thereby positioning its contribution more precisely within the broader context of representation learning for NAS.

- It would be useful to evaluate whether similar gains can be achieved with alternative encoders such as BERT, RoBERTa, T5, or even randomly initialized or domain-specific text encoders. Without such a comparison, it is difficult to isolate the specific contribution of CLIP’s multimodal pretraining. A systematic ablation varying the text encoder would also clarify whether CLUE-NAS’s performance stems from the general use of a language model or from CLIP’s unique image–text alignment capabilities.

- Another weakness of the paper is the absence of the source code.

[1] Chatzianastasis, Michail, et al. "Graph-based neural architecture search with operation embeddings." Proceedings of the IEEE/CVF International Conference on Computer Vision. 2021.

**Questions:**

- Have the authors explored alternative integration strategies, such as using CLIP embeddings as node-level features within the graph encoder, without the contrastive learning loss?

- Have the authors conducted or considered an ablation with alternative text encoders, such as BERT, RoBERTa, T5, or a randomly initialized transformer? Would similar gains be expected if CLUE-NAS were trained with purely textual models lacking image–text alignment?

- Could the authors comment on how their approach relates to prior work in this area, such as the operation embedding methods?

- Have the authors tested whether alternative or more detailed prompts (e.g., including hierarchical structure, layer types, or depth information) affect the resulting embeddings or downstream prediction performance?

---

### Meta-Review · Area_Chair_h7Ln · 2025-12-06

**Summary:**

This paper proposes CLUE-NAS, an encoder-based performance predictor for Neural Architecture Search (NAS) that integrates structural graph-based representations with semantic embeddings derived from CLIP’s text encoder. The method aligns graph embeddings with natural-language–based architecture descriptions via contrastive learning, with the goal of enhancing generalization across search spaces and improving prediction in low-data regimes. Experiments are conducted on NAS-Bench-101, 201, and 301.

Most reviewers agree that the idea of incorporating CLIP-based semantic context into NAS encoders is interesting. However, all reviewers raised major concerns, including weak experimental depth, insufficient comparisons, unclear novelty relative to prior work, and missing ablations. For example, Reviewer s1Ro raised concern about novelty relative to prior generalized encoders such as CDP. Reviewer MFSo pointed out the lack of exploration of alternative integration strategies. No results on large-scale benchmarks such as ImageNet-1K and NAS benchmarks besides image classification, raised by reviewer 98k1 and s1Ro. Lack of clarification of search cost fairness raised by reviewer 98k1.

All the reviewers reach a consensus on rejection. The authors did not provide a rebuttal, so these issues remained completely unresolved.

**Reviewer Concerns:**

The authors did not participate in the rebuttal, so none of the reviewers’ concerns were addressed.

**Reviewer Scores:**

Given the lack of author response, reviewers would not increase their scores. So the scores remain as follows:

* Reviewer MFSo (Rating: 4)
* Reviewer 98k1 (Rating: 2)
* Reviewer s1Ro (Rating: 2)

---

### Decision · Program_Chairs · 2026-01-26

Reject